# Tumor-Infiltrating Lymphocytes and Immune Response in HER2-Positive Breast Cancer

**DOI:** 10.3390/cancers14246034

**Published:** 2022-12-08

**Authors:** Melani Luque, Marta Sanz-Álvarez, Miriam Morales-Gallego, Juan Madoz-Gúrpide, Sandra Zazo, Carolina Domínguez, Alicia Cazorla, Yann Izarzugaza, Juan Luis Arranz, Ion Cristóbal, Federico Rojo

**Affiliations:** 1Department of Pathology, Fundación Jiménez Díaz University Hospital Health Research Institute (IIS—FJD, UAM)—CIBERONC, 28040 Madrid, Spain; 2Medical Oncology Department, Fundación Jiménez Díaz University Hospital, 28040 Madrid, Spain; 3Cancer Unit for Research on Novel Therapeutic Targets, Oncohealth Institute, IIS-Fundación Jiménez Díaz-UAM, 28040 Madrid, Spain; 4Translational Oncology Division, Fundación Jiménez Díaz University Hospital Health Research Institute, UAM, 28040 Madrid, Spain

**Keywords:** HER2-positive breast cancer, immunotherapy, tumor-infiltrating lymphocytes

## Abstract

**Simple Summary:**

It has been known for decades that the immune system plays an important role in the etiology of breast cancer. Also, lymph node spread is the most important prognostic factor in breast cancer, and the presence of tumor-infiltrating lymphocytes (TILs) predicts a beneficial anti-HER2 therapeutic response. The latest translational clinical research aims to strengthen a patient’s immune system to tackle and kill cancer cells more effectively. However, immune system cells can either establish a protective antitumor response or, conversely, induce chronic inflammation that promotes disease progression. This ambivalence depends, to a large extent, on the immune cell infiltrate present in the tumor and the communication that these cells establish with the tumor cells. This review aims to summarize the current knowledge of the immune system–breast cancer relationship, emphasizing TILs and their importance as biomarkers of clinical progression of the disease.

**Abstract:**

Human epidermal growth factor receptor 2–positive (HER2-positive) breast cancer accounts for 15 to 25% of breast cancer cases. Although therapies based on the use of monoclonal anti-HER2 antibodies present clinical benefit for a subtype of patients with HER2-positive breast cancer, more than 50% of them are unresponsive to targeted therapies or they eventually relapse. In recent years, reactivation of the adaptive immune system in patients with solid tumors has emerged as a therapeutic option with great potential for clinical benefit. Since the approval of the first treatment directed against HER2 as a therapeutic target, the range of clinical options has expanded greatly, and, in this sense, cellular immunotherapy with T cells relies on the cytotoxicity generated by these cells, which ultimately leads to antitumor activity. Lymphocytic infiltration of tumors encompasses a heterogeneous population of immune cells within the tumor microenvironment that exhibits distinct patterns of immune activation and exhaustion. The prevalence and prognostic value of tumor-infiltrating lymphocyte (TIL) counts are associated with a favorable prognosis in HER2-positive breast cancers. This review discusses emerging findings that contribute to a better understanding of the role of immune infiltrates in HER2-positive breast cancer. In addition, it summarizes the most recent results in HER2-positive breast cancer immunotherapy and anticipates which therapeutic strategies could be applied in the immediate future.

## 1. The Importance of Immune Surveillance in HER2-Positive Breast Cancer

Breast cancer represents about 30% of all cancer cases among women, being the most commonly diagnosed cancer type and one of the leading causes of cancer death [1]. Breast cancer is a highly heterogeneous disease that encompasses multiple subtypes, with diverse morphological and clinical features [2]. Amplification and/or overexpression of HER2 confers high aggressiveness and the HER2-positive molecular subtype represents around 20% of all breast cancers patients [3]. Thus, HER2-blocking antibodies, such as trastuzumab and pertuzumab, in combination with chemotherapy represent the standard therapy for HER2-positive patients [4]. The antibody drug trastuzumab-emtansine (T-DM1) has also been approved for the treatment of advanced HER2-positive patients with progressive disease after the treatment with the first-line therapy mentioned [5]. Trastuzumab-deruxtecan (T-DXd), another antibody–drug conjugate, has recently been demonstrated to improve progression-free survival (PFS) rates compared with T-DM1 in metastatic HER2-positive patients, and it is currently evaluated to be positioned as a second-line therapy in those patients [6,7]. However, about 25–30% of HER2-positive tumors finally relapse, as a result of primary and secondary resistance [8,9,10]. Thus, the development of new therapeutic approaches to overcome therapy resistance still remains a high-priority issue.

As compared with other carcinomas, breast tumors have a low mutation burden, therefore being traditionally considered a poorly immunogenic cancer [11]. However, a great body of evidence has shown the existence of remarkable heterogeneity, with triple negative (TN) and HER2-positive having the highest immunogenic potentials within the molecular subtypes of breast cancer [12,13].

In the context of cancer immunology research, the understanding of the crosstalk between tumor and the host’s immune cells remains one of the most challenging issues. It is necessary to delve deeper into the mechanisms involved in the regulation of disease progression by the immune system in order to design more effective immunotherapies that improve clinical outcomes. In this regard, lymphocytes (including T, B, and Natural Killer (NK) cells), as well as macrophages, are the key players of the immune response triggered against tumor cells [14]. Some tumor cells manage to evade the organism’s intrinsic tumor-suppressive mechanisms. However, they can be destroyed by the host immune system before they can establish malignancy and progress. Through the production of interferon-γ (INF-γ), a cascade of responses is originated by NK cells, such as the recruitment of more NK cells and macrophages (innate immunity). This killing process also involves CD4+ and CD8+ effector T cells, which are capable of recognizing and eliminating transformed cells [15,16,17]. Even so, cancer cells can survive immune destruction and may enter a state of tumor dormancy or an equilibrium phase. In this process, INF-γ and lymphocytes exert a selection pressure that stimulates the appearance of mutations in some tumor cells that allow them to acquire abilities to resist immune attack. It remains to be determined which factors contribute to tip the balance either in the direction of cellular elimination or immune escape. In this regard, in a clinical context, some studies seem to indicate that tumors can stay in patients in a state of dormancy for years [16,18]. In the process of escape, surviving tumor variants begin to progressively expand, establishing an immunosuppressive tumor microenvironment and eventually generating clinically detectable cancer entities [15,19].

## 2. Tumor-Infiltrating Lymphocytes in HER2-Positive Breast Cancer

Tumor-infiltrating lymphocytes (TILs) are mononuclear immune cells that infiltrate tumor tissue and have been described in most types of solid tumors, such as breast cancer, colon cancer, melanoma, and lung cancer [20,21,22]. TILs comprise CD8+ and CD4+ effector T cells, regulatory T cells (Treg), B cells, NK cells, and macrophages, and not only the amount of lymphocytic infiltration, but also the different immune populations determine the clinical outcome. TILs are responsible for the local immune response directed against cancer cells to prevent tumor growth and metastasis. In recent years, the evaluation of TILs has progressively gained a great deal of attention, as a large body of evidence has accumulated supporting their clinical validity as a tumor biomarker in cancer, including breast cancer [23]. In particular, those cases with at least 50–60% lymphocytic infiltration have been defined as lymphocyte-predominant breast cancer (LPBC) and have been reported to have better outcomes compared to those with less lymphocytic infiltration, highlighting that it has the potential to serve as an alternative therapeutic target in this disease [24].

In response to the increasing clinical significance of TILs in breast cancer, an international working group on immuno-oncology biomarkers was developed to set up and define a common methodology for a homogeneous evaluation of TILs in breast cancer samples. Commonly, the presence of TILs is studied using slides stained with hematoxylin and eosin (H&E) and analyzed by light microscopy, where they can be identified by their morphological characteristics or additional lymphocytic markers [25,26]. TILs have an intratumoral or stromal location. Thus, lymphocytes having direct contact with tumor cells are defined as intratumoral TILs, whereas those lymphocytes located throughout the stroma without direct contact with carcinoma cells are defined as stromal TILs. As both are located in the tumor tissue region, they are considered “true” TILs. Intratumoral TILs were initially considered more relevant and, therefore, more useful for diagnosis, but most recent studies point to stromal TILs as a more valuable and reproducible parameter. It is usually difficult to observe intratumoral TILs on H&E-stained slides due to their heterogeneity, low numbers, and fewer cases. Therefore, the current recommendation of the TIL working group is to assess mainly stromal TILs [25].

Breast cancer TILs primarily contain CD8+ and CD4+ T cells, with smaller amounts of regulatory T cells (Treg), B cells, NK cells, and macrophages [27,28]. Different subtypes of breast cancer exhibit different levels of TIL infiltration, with TN and HER2-positive subgroups typically showing greater infiltration than luminal subtypes [26]. TN tumors have been found to show a prevalence of cases with LPBC up to 20%, followed by the HER2-positive subtype with 16% of LPBCs, whereas the luminal subgroup has the lowest prevalence (6%) [24].

In the last decade, several studies—including clinical trials—have evaluated the relevance of TILs in early breast cancer (Table 1). A systematic review by Mao and colleagues evaluated TIL levels in biopsies obtained before neoadjuvant chemotherapy. They reported a significant association between high TIL numbers and increased pathological complete response (pCR) after treatment. Furthermore, TILs showed predicted values of pCR only in TN and HER2-positive breast cancer patients [29]. In the GeparSixto trial, the correlations among pCR rates, high TILs, and LPBC were examined in breast cancer patients who received neoadjuvant chemotherapy (Table 1). Stromal TILs, as well as LPBC, were significantly associated with pCR in univariable and multivariable analyses in HER2-positive tumors. In HER2-positive patients, the addition of carboplatin produced an increase in pCR rates in LPBC tumors relative to patients receiving anthracycline plus taxane treatment alone [30]. Interestingly, Denkter and colleagues performed an analysis including 3771 breast cancer patients treated with neoadjuvant chemotherapy. As expected, they observed higher prevalence of TILs in both the TN (30%) and HER2-positive (20%) subtypes compared with the luminal subtype (13%). Across all molecular subtypes, high TIL levels were associated with responses to neoadjuvant chemotherapy. However, an association between longer overall survival and higher levels of TILs in the HER2-positive subtype was not identified [31]. 

The FinHER trial compared 9 weeks of trastuzumab plus chemotherapy versus chemotherapy alone in HER2-positive patients [33]. They reported a significant correlation between high TIL numbers and greater clinical benefits with the trastuzumab addition (Table 1). They also determined a reduction in distant recurrence risk for each 10% increase in TILs only among patients treated with trastuzumab [34]. Furthermore, in the APHINITY trial, 4805 HER2-positive breast cancer patients were randomly assigned to either pertuzumab or placebo with adjuvant chemotherapy plus trastuzumab [35] (Table 1). Adding pertuzumab to the regimen was found to improve prognosis for patients with higher TIL levels [36]. In the N9831 trial, patients treated with chemotherapy alone, but not with trastuzumab, showed significant improvement in relapse-free survival (RFS) when higher stromal TIL levels were identified [38] (Table 1). In the same way, the NRG/NSABP B-31 trial evaluated early HER2-positive breast cancer patients who received regimens of chemotherapy with or without trastuzumab. The results did not show differences in either therapeutic arm in terms of improved disease-free survival (DFS) [39]. In the NeoALTTO trial, TIL levels were identified as a prognostic factor for both pCR and event-free survival (EFS) in early HER2-positive patients treated with lapatinib plus trastuzumab [41] (Table 1). Additionally, in the ShortHER trial authors compared 9 weeks versus 1 year of trastuzumab treatment in addition to chemotherapy. Interestingly, it was observed that 10% TIL increments were positively and independently associated with a 27% reduction in the risk of distant DFS (Table 1). Patients with low levels of TILs were also found to benefit more from long-term trastuzumab treatment [43].

Despite the well-established prognostic implications of TILs in early disease, their role in advanced disease is less clear. In the Cleopatra trial, TIL values were evaluated in advanced HER2-positive patients who received pertuzumab or placebo in addition to trastuzumab and docetaxel. They observed a significant association between higher TIL infiltration and improved OS [45] (Table 1). A total of 652 patients with metastatic HER2-positive breast cancer were enrolled in the MA.31 trial. Patients had not been previously treated with chemotherapy or anti-HER2 agents in the metastatic setting and were randomized to receive taxane-based chemotherapy in combination with trastuzumab or lapatinib [46] (Table 1). Patients treated with lapatinib showed worse prognoses when they had low CD8+ TIL levels compared to patients treated with trastuzumab [47]. Altogether, the results of these two studies performed in advanced HER2-positive disease indicate that TIL counts could apparently have a smaller prognostic significance in the metastatic setting compared to early disease.

## 3. Clinical Relevance of Different Phenotypes of Immune Infiltrates

Besides the mere count of lymphocytic infiltration, the phenotype of the lymphocytes may also dictate the clinical outcome of HER2-positive breast cancer patients, as every specific subset has a specific role in cancer development. Cumulative data from human studies have associated the different immune populations with a predominant contribution to either pro- or antitumor activities (Figure 1).

### 3.1. CD4+ and CD8+ T Cells

T cells, including CD4+ and CD8+ cells, function as immune effectors to induce adaptive immunity. After activation, CD8+ T lymphocytes differentiate into cytotoxic cells, and CD4+ T cells originate three subpopulations of T helper (Th) cells: type 1 Th (Th1), type 2 Th (Th2), and type 17 Th (Th17) cells. CD8 lymphocytes destroy tumor cells and CD4+ subpopulations can produce pro- or antitumor responses [48,49]. CD4+ Th-1 cells release proinflammatory cytokines, such as TNF-α, IL-2, and INF-γ, inducing a potent antitumor response. Further, they promote the antitumor activity of macrophages, including NK and T cells [49,50]. CD4+ Th-2 cells secrete cytokines that have been reported to play a role in tumor growth and metastasis, such as IL-4, IL-5, and IL-13 [51]. However, they also release IL-10, which has both pro- and antitumor properties [52]. Additionally, CD4+ Th-17 cells can induce tumor growth after being activated by TGF-β, IL-6, or IL-23 [53].

In HER2-positive breast cancer, a variety of studies have evaluated the significance of T cell subpopulations on the onset and progression of the disease. In a cohort of 1334 primary invasive breast cancer patients with long-term follow-up, CD8+ T lymphocyte density was assessed. The results showed a significant association between a higher number of CD8+ lymphocytes and a better clinical outcome, regardless of other clinical parameters, including HER2 status [54]. Moreover, elevated CD8+ and low Forkhead box protein 3 positive (FOXP3+) T-cell infiltrates have been identified to serve as an independent predictor of improved OS and RFS after treatment with neoadjuvant chemotherapy in patients with HER2-positive and HER2-negative breast cancer [55]. In addition, breast carcinomas with a higher number of CD8+ T cells have shown a greater benefit from treatment with trastuzumab [56]. Recently, prognostic subsets of T cells have been identified in breast tumors by an automated tool to determine optimal cluster numbers in single-cell RNA sequencing data. Consistent with previous results, CD8+ subsets and Tregs were associated with improved survival in breast cancer patients, and a significant correlation between CD4+ naive T cell expression and OS was identified in breast cancer subtypes HER2-positive and TN [57]. Lately, patients with early HER2-positive breast cancer treated with anti-HER2 therapies, lapatinib plus trastuzumab, in the absence of chemotherapy, have shown a correlation between TIL subsets and pCR rate. The results showed higher pCR rates in patients with CD4+, CD8+, high CD20+ stromal TILs, and high CD20+ intratumoral TILs who had been treated with anti-HER2 therapies [58]. In addition, Datta and colleagues evaluated the CD4+ Th1 immune response in patients with primary invasive HER2-positive breast cancer treated with trastuzumab plus chemotherapy. The results showed that patients who achieved pCR after receiving neoadjuvant therapy had a greater Th1 CD4+ response [59].

### 3.2. Treg Cells

Treg cells were described in 1995 and comprise a distinct group of CD4+ T lymphocytes with immunosuppressive properties [60]. Treg cell activity has been related to poor immunological response, and it has been proposed that they could represent a critical mechanism for immune evasion by tumors, including breast cancers. Treg cells are characterized by the expression of FOXP3, a transcription factor that participates in the differentiation, development, maintenance, and function of the Treg cell population [61]. The suppression of the immune response by Treg cells is produced by different mechanisms. These include the inhibition of activated T cells through interaction with the CTLA-4 protein that inhibits the costimulatory molecule of CD28 T cells, as well as the release of cytokines that promote the death of tumor T cells, such as TGF-β, IL-10, and IL-35 [62].

The role of Treg cells in HER2-positive breast cancer has been evaluated in several studies. In a cohort of 3992 breast cancer patients, Treg TILs were assessed by immunohistochemistry. They identified an improved survival in HER2-positive/ER (estrogen receptor)-negative patients with high Treg TIL infiltrates and coexistent CD8+ T-cells. Nevertheless, high Treg TIL quantities without CD8+ T cells were not associated with better survival in ER-positive breast cancer patients [63]. Based on the studies to date, further research is needed to clarify the role of Treg cells in tumoral immune response and their interaction with other cell populations present in the tumor [64]. Thus, future studies on their prognostic role in patient samples are necessary to finally determine the clinical importance of Treg cell assessment.

### 3.3. B Cells

In contrast to the high attention given to T lymphocytes in tumor progression, the role of B lymphocytes within the tumor microenvironment remains underexplored. In addition to their function of secretion of antibodies and cytokines, which trigger a humoral antitumor response, B cells are able to recognize antigens, regulate antigen processing and presentation, and promote and modulate innate and T cell immunity [65]. This contributes to a plethora of functions in the tumor microenvironment, sometimes contradictory between promotion and regression [66]. Finally, increasing evidence has revealed that high B cell levels are a positive prognostic factor in breast cancer. In fact, tumor-infiltrating B cell densities have been shown to increase in breast tumor tissue compared with normal breast tissue. Moreover, tumor-infiltrating B cells have also been associated with global TILs, CD4+ and CD8+ T cells, higher tumor grade and proliferation, and HR negativity [67]. In a cohort of biopsies from invasive breast cancer patients obtained before neoadjuvant chemotherapy treatment, the expression of CD3, CD8, and CD20 markers in stromal TIL infiltrates was evaluated. Interestingly, they identified CD20 as the most sensitive and specific marker predicting pCR after chemotherapy treatment. Specifically, high CD20 expression was also described to predict pCR in those cases with the HER2-positive subtype [68]. In HER2-positive breast cancer, high levels of B lymphocytes has been correlated with a higher proportion of patients achieving pCR, following lapatinib and trastuzumab treatment without chemotherapy [58]. In contrast, another study showed higher levels of tumor-infiltrating B cells in the pretreatment biopsies of breast cancer patients. Following neoadjuvant treatment with anthracycline plus taxane-based therapy or with trastuzumab, a statistically significant decrease in B cell counts in tumor samples was reported [69]. Therefore, additional research is required to determine the clinical relevance of this immune population in breast cancer due to considerable controversy regarding the prognostic impact of this subpopulation of immune cells.

### 3.4. NK Cells

NK cells are the main players of innate immunity. Although they are cytotoxic effector cells, they also are involved in the modulation of immune reactions through the secretion of cytokines and chemokines. They are identified by their surface markers CD56 and CD16 and can be subdivided into different populations according to their relative expression [70]. Since many studies have demonstrated their cytotoxicity against tumoral cells, NK cells are recognized as crucial agents of immunosurveillance and elimination phases during cancer immunoediting [71]. However, their specific role in breast tumor development, as well as in therapy response, remains still unclear.

A study performed with early breast cancer patients showed the association between the increased expression of molecules involved in the activation of NK cells (such as NK cell receptors) and the increase in RFS [72]. In patients with locally advanced disease, high levels of NK cells were found to be significantly related with an increase in pCR rates. Further, the reduction in NK-mediated cytotoxicity was significantly associated with a poorer response to neoadjuvant chemotherapy [73]. Particularly in patients with HER2-positive breast cancer that had been treated with anti-HER2 agents, an association between increased NK cell levels and pCR was observed [74]. Moreover, in vivo studies using HER2-positive breast cancer mouse models have shown that NK cell depletion abolished the activity of anti-HER2 monoclonal antibodies [75,76]. In patients with early breast cancer, HER2-specific antibodies have been shown to trigger NK cell-mediated antibody-dependent cellular cytotoxicity (ADCC) [77,78]. New therapeutic approaches aiming to enhance NK cell activation by anti-HER2 agents are currently being evaluated [79], including the systemic treatment with recombinant cytokines [80], ADAM inhibitors [81], and the delivering of Toll-like receptor (TLR) ligands to the tumor site [82].

It has been suggested that the efficacy of HER2-blocking agents could be affected by NK cell differentiation [83]. Interestingly, circulating CD57+ NK cell numbers have been found to be associated with the acquisition of resistance to anti-HER2 therapies [83]. Therefore, CD57+ NK cell determination could emerge as a useful biomarker for improving clinical management of these sets of patients.

### 3.5. Tumor-Associated Macrophages

Many immune cells are key players in the tumor microenvironment. However, tumor-associated macrophages (TAMs) have gained special attention in the last decade. In addition to their ability to phagocytize cancer cells [84], they can also recruit other immune cells, as well as present antigens to T cells [85]. Macrophages are commonly grouped into M1 and M2 macrophages, which are two polarized groups discriminated by different functions and cell surface markers. However, both M1 and M2 tumor-infiltrating macrophages are generally identified by the CD68 marker [84,85]. Over the years, several mechanisms by which TAMs can modulate tumor microenvironment have been described, including immunosuppressive actions through programmed death-ligand 1 (PD-L1) expression and the promotion of tumor growth, invasion, and angiogenesis [86,87,88]. M1 macrophages undergo classical macrophage activation, which means they are stimulated by INF-γ and TLR ligands. The M1 phenotype is characterized by expression of CD80 and CD86 (costimulatory molecules) and the secretion of cytokines with proinflammatory properties (such as IL-12 and IL-23). They are commonly associated with Th1 cytotoxic responses that promote tumor destruction [89,90]. M2 macrophages have an alternative activation, stimulated by IL-4/IL-13. They are characterized by CD163, CD204, and CD206 markers, the secretion of IL-10, and the expression of EGF and VEGF. They have been typically associated with immunosuppressive and protumorigenic effects because of their contribution to the activation of Th2 immune responses [90,91,92]. As a result of the accumulating evidence enhancing their role in tumor development, TAMs have emerged as a promising therapeutic target in breast cancer research [93,94]. However, their role is not fully understood, and their relationship with therapy efficacy and patient outcome remains unclear.

Since elevated TAM (CD68+) infiltration in breast stroma has been negatively correlated with clinic-pathological features, such as tumor size, HR status, histologic grade, and age [95,96], it has been disclosed as a prognosis factor in breast cancer patients [97]. Of note, it could be difficult to deepen the interpretation of these results because of the fact that CD68, despite being a pan-macrophage marker, may be expressed in other immune populations [98].

Studies in early breast cancer patients have shown that high numbers of CD163+ M2-macrophages significantly correlate with short RFS and OS rates [99,100]. In this set of patients, high CD163+ infiltration was strongly associated with unfavorable prognosis factors, such as proliferation, poor tumor differentiation, and ER negativity, supporting the importance of TAM polarization into M1 and M2 phenotypes in breast cancer disease [99]. In human invasive breast cancer samples, a high density of TAMs was found to be significantly correlated with the reduction of RFS and OS. Further, TAM infiltration was reported as an independent prognostic factor [101]. T-cell immunoglobulin and mucin domain-3 (TIM-3) is an immune checkpoint molecule that may be expressed in T cells. Interestingly, TIM-3 may represent a potential target to overcome resistance to programmed cell death protein 1 (PD-1) blockade therapies [102]. In HER2-positive patients with metastatic disease (with brain metastases), high densities of TIM-3+ CD163+ macrophages have recently been associated with worse OS of patients [103]. Interestingly, TAM polarization and recruitment into the tumor microenvironment have been associated, not only with poor clinical outcomes in patients with breast tumors [100,104,105], but also with the hampering of anti-HER2-specific agents’ activity [106]. As mentioned previously, M2 macrophage phenotypes have been highly associated with tumoral progression. Importantly, in mouse models, TAM depletion has been shown to improve the therapeutic effects of the anti-HER2 antibody trastuzumab [107]. In addition, the study also highlights the relevance of the repolarization of M2 TAMs into M1 phenotypes after treatment with IL-21, as it has also been reported to enhance the therapeutic effect of trastuzumab [107]. Although these findings present a promising approach to address resistance, further research will be needed to clarify the potential value of TAM repolarization in clinical practice.

## 4. Immunotherapy in HER2-Positive Breast Cancer

In recent times, the ability of clinical agents to influence the body’s own ability to recognize and attack cancer cells, orchestrating treatment through the immune system, has been gaining traction in immunotherapy as an innovative antitumor strategy. Immunotherapy comprises a broad variety of therapeutic strategies, including immune checkpoint inhibitors, monoclonal and bispecific antibodies, vaccines, and antibody–drug conjugates. As mentioned above, HER2-positive and TN breast cancer subtypes are sufficiently immunogenic to be considered potential candidates for immunotherapy [108,109,110].

The immune checkpoint inhibitors atezolizumab and pembrolizumab, both PD-1/PD-L1 axis inhibitors, have been recently approved by the FDA for the treatment of unresectable, locally advanced or metastatic TN breast cancer [111,112]. As a result, PD-L1 expression assessment has emerged as a biomarker to guide immunotherapy treatment in breast cancer. The PD-L1 antigen on tumor cells interacts with PD-1 on the surface of the cytotoxic CD8+ T cells to inhibit their activity and stimulate Treg cell development, thereby suppressing the immune response [113]. Although the initial data have shown that the greatest benefit is achieved in the TN breast cancer subtype, PD-L1 positivity is occasionally found in all metastatic breast cancer subtypes, including the HER2-positive subgroup [114]. In this respect, in the PANACEA trial, trastuzumab-resistant patients with advanced HER2-positive breast cancer were randomized to receive pembrolizumab plus trastuzumab. Combined treatment showed greater clinical benefit in patients with PD-L1+ tumors. Moreover, they reported higher TIL levels in PD-L1+ tumors [115]. Additionally, HER2-enriched breast cancer patients were randomized to receive trastuzumab plus pertuzumab in combination with pembrolizumab in the absence of chemotherapy in the prospective phase II KEYRICHED-1 trial. The results of this study showed comparable pCR rates after immunotherapy alone or in combination with chemotherapy. This means that an appropriate molecular selection of patients might achieve clinically meaningful pCR rates, similar to those obtained with longer and more toxic chemotherapy regimens [114]. In the KATE2 trial, patients with advanced HER2-positive breast cancer treated with atezolizumab plus TDM-1 did not show an increase in PFS and showed more adverse events [116,117]. The phase III IMpassion050 trial evaluated the addition of atezolizumab to neoadjuvant anti-HER2 therapy in early HER2-positive breast cancer patients. The results did not show an improvement in pCR rates after addition of anti-PD-1 inhibitor [118]. To conclude, future studies will be necessary to determine the possible benefit of immunotherapies in additional subgroups of patients with advanced breast cancer enriched for the expression of PD-L1.

## 5. Conclusions

In recent years, a growing body of evidence has emerged highlighting the role of immune infiltrates in breast tumors. In the HER2-positive subtype, the assessment of TILs in general has become a predictive and prognostic biomarker of response to systemic and anti-HER2 therapies. However, other components of breast cancer immune infiltrates, including specific T-cell subsets, NK cells, and TAMs, need further investigation to determine their specific roles and relevance in the clinical management of HER2-positive patients. Despite this lack of robustness in biological knowledge, recent discoveries in immunotherapy have progressively changed the landscape of cancer treatment. Although HER2-positive breast cancer is identified as an immunogenic carcinoma, to date, immunotherapies have provided mild therapeutic effects. Therefore, further research is needed to clarify the contribution and clinical value of different immune subpopulations before these therapies provide greater clinical benefit for HER2-positive breast cancer patients.

## Figures and Tables

**Figure 1 cancers-14-06034-f001:**
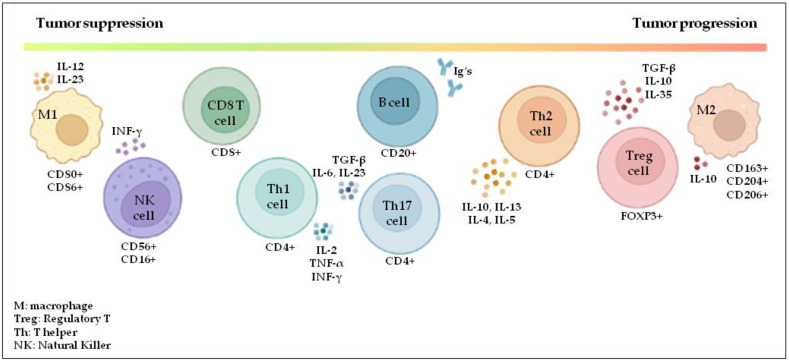
The predominant contribution of the different lymphocytic infiltrates to either tumor suppression or progression, including NK cells, cytotoxic T cells, T helper (Th) cell subsets, regulatory T (Treg) cells, B cells, and tumor-associated macrophages, comprising those with antitumorigenic (M1) or protumorigenic (M2) properties. Through the secretion of different factors, these immune populations play key roles in shaping the microenvironment, thereby driving either immune-mediated anti- or protumor activity. (Created with BioRender.com, accessed on 17 January 2022).

**Table 1 cancers-14-06034-t001:** Summary of main clinical trials with TIL evaluation in HER2-positive breast cancer.

Trial	Reference (Clinical Trial)	Setting	Number of Patients	Treatment Arms	Parameters	Results	*p*-Value	Reference (TILs Evaluation)
GeparSixto	[32]	Neoadjuvant	580	Paclitaxel/ doxorubicin/trastuzumab Paclitaxel/ doxorubicin/trastuzumab/carboplatin	Association between TILs (as a continuous parameter) or LPBC (>60%) and pCR rates	10% increase in TILs was significantly associated with pCR. sTILs and LPBC were independent predictor factors for pCR. LPBC tumors treated with PMCb showed high-response pCR rates.	0.001 0.001; 0.001 0.006	[30]
FinHER	[33]	Early	232	Docetaxel/ FEC Docetaxel/ FEC/ Trastuzumab	Association between TILs (as a continuous variable) and trastuzumab benefit	10% increase in TILs was significantly associated with reduction in DDFS.	0.025	[34]
APHINITY	[35]	Early	4804	Chemotherapy/ trastuzumab Chemotherapy/ trastuzumab/ pertuzumab	Association between TILs (as a continuous variable) and 6-year IDFS after addition of Pertuzumab	TILs percentage appeared to be more predictive of pertuzumab treatment effect than clinical composite risk score.	n.s.	[36]
N9831	[37]	Early	945	Doxorubicin/ cyclophosphamide/ paclitaxel (Arm A) Doxorubicin/ cyclophosphamide/ paclitaxel/ trastuzumab (Arm C)	Association between sTILs (>60%) and RFS after addition of trastuzumab	Patients with high and low sTILs did not show differences in 10-year RFS.	0.63	[38]
NRG/NSABP B-31	[37]	Early	1581	Chemotherapy Chemotherapy/ Trastuzumab	Association of sTILs (as a semi-continuous variable) and trastuzumab benefit in DFS	Increase in sTILs was significantly associated with improved DFS. There was not association between sTILs and trastuzumab benefit.	0.001 0.65	[39]
NeoALTTO	[40]	Early	387	Chemotherapy/ trastuzumab Chemotherapy/ lapatinib Chemotherapy/ trastuzumab/ lapatinib	Associations among presence of TILs (>5%), pCR, and EFS	Levels of TILs greater than 5% were associated with higher pCR rates. Every 1% increase in TILs was associated with a decrease in EFS.	0.01 0.002	[41]
ShortHER	[42]	Early	866	Anthracycline/ taxane/ trastuzumab 1 year (Arm A) Anthracycline/ taxane/ trastuzumab 9 weeks (Arm B)	Association between TILs (as a semicontinuous and binary variable) with DDFS	10% increase in TILs was an independent prognostic factor for DDFS. 10% increase in TILs was a significant prognostic factor in arm B Arm A patients with <20% TILs showed better DDFS compared with arm B patients.	0.002 0.009; 0.021	[43]
CLEOPATRA	[44]	Advanced	678	Docetaxel/ trastuzumab Docetaxel/ trastuzumab/ pertuzumab	Association between sTILs (as a semicontinuous variable) and PFS, OS and pertuzumab benefit	There was not significant association between TILs and PFS. 10% increase in sTILs was significantly associated with better OS. There were not significant differences by sTILs for PFS or OS after pertuzumab addition.	0.063 0.0014 0.23; 0.21	[45]
MA.31	[46]	Advanced	427	Taxane/ trastuzumab Taxane/lapatinib	Association between TILs (>5%) and PFS after addition of lapatinib	Low CD8+ cytotoxic sTILs were associated with worse PFS in patients treated with lapatinib.	0.03	[47]

TILs: tumor-infiltrating lymphocytes; LPBC: lymphocyte-predominant breast cancer; pCR: pathological complete response; sTILs: stromal TILs; PMCb: Paclitaxel plus no pegylated liposomal doxorubicin (Myocet^®^) plus Carboplatin; FEC: fluorouracil, epirubicin, and cyclophosphamide; DDFS: distant disease-free survival; IDFS: invasive disease-free survival; n.s.: no significance; RFS: relapse-free survival; DFS: disease-free survival; EFS: event-free survival; PFS: progression-free survival; OS: overall survival.

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
