# Peer review of "Tumor-Infiltrating Lymphocytes and Immune Response in HER2-Positive Breast Cancer"

_cancers, 2022, doi:10.3390/cancers14246034_

Round 1

Reviewer 1 Report

The review reads is comprehensive and reads well.

It seams that no one has checked the manuscript after adding references.

All the refs number should be checked.

Many of the numbers include dots before and after the number, some of them do not have an appropriate space.

There is no consistency in the way the refs numbers are presented.

Severa; content mistakes:

Marophages are not type of lymphocytes- line 77

Atezolizumab is a PDL1 inh line 371

"pDL-1 positivity is relatively common - This is wrong , line 378 

Many word lack appropriate space between for example: line 69, ;on2 112, lin3 128

Author Response

RESPONSE TO REVIEWER 1 COMMENTS

Comments and Suggestions for Authors

The review reads is comprehensive and reads well.

It seams that no one has checked the manuscript after adding references.

All the refs number should be checked.

Many of the numbers include dots before and after the number, some of them do not have an appropriate space.

There is no consistency in the way the refs numbers are presented.

We welcome your comments. The document has been thoroughly reviewed for references and formatting errors.

Severa; content mistakes:

Marophages are not type of lymphocytes- line 77.

Atezolizumab is a PDL1 inh line 371.

"pDL-1 positivity is relatively common - This is wrong , line 378.

Thank you for your corrections. The sentences have been rewritten to correct the errors you pointed out.

Many word lack appropriate space between for example: line 69, ;on2 112, lin3 128.

The document has been thoroughly reviewed for typos.

Reviewer 2 Report

The review discusses relevant data in the understanding of the role of immune infiltrates in HER2-positive breast cancer, and discuss the prevalence and prognostic value of tumor-infiltrating lymphocytes (TIL) counts in HER2-positive breast cancers. Furthermore, it summarizes the most recent results in HER2-positive breast cancer immunotherapy and discuss therapeutic strategies that could be applied in the immediate future. This type of analysis if of high interest in clinical settings and has a high value in order to optimize therapy, decrease toxicities and improve quality of life of patients.

The authors conclude that in HER2-positive subtype, assessment of TILs could be used as a predictive and prognostic biomarker of response to systemic and anti-HER2 therapies. Also, the authors point out as of high interest the study of immune infiltrates, including specific T-cell subsets, NK cells and TAMs in breast cancer. These conclusions have been supported by the data analysis of trials and discussion.

In order to improve the article is highly simplify the data in Table 1, for example classify the trials according to the pharmacological treatment and setting. Add also the number of patients, please simplify and unify how the results are explained in the table (for example, indicate the parameter 10% TILS…). Authors should edit the text through the article, for example see line 183.

Author Response

RESPONSE TO REVIEWER 2 COMMENTS

Comments and Suggestions for Authors

The review discusses relevant data in the understanding of the role of immune infiltrates in HER2-positive breast cancer, and discuss the prevalence and prognostic value of tumor-infiltrating lymphocytes (TIL) counts in HER2-positive breast cancers. Furthermore, it summarizes the most recent results in HER2-positive breast cancer immunotherapy and discuss therapeutic strategies that could be applied in the immediate future. This type of analysis if of high interest in clinical settings and has a high value in order to optimize therapy, decrease toxicities and improve quality of life of patients.

The authors conclude that in HER2-positive subtype, assessment of TILs could be used as a predictive and prognostic biomarker of response to systemic and anti-HER2 therapies. Also, the authors point out as of high interest the study of immune infiltrates, including specific T-cell subsets, NK cells and TAMs in breast cancer. These conclusions have been supported by the data analysis of trials and discussion.

In order to improve the article is highly simplify the data in Table 1, for example classify the trials according to the pharmacological treatment and setting. Add also the number of patients, please simplify and unify how the results are explained in the table (for example, indicate the parameter 10% TILS…).

Thank you for your comments and observations. In accordance with them, we have included in Table 1 a new column showing the number of patients in each trial. In addition, as you suggested, we have unified the results whenever possible and simplified them for easier understanding.  On the other hand, although we have tried to classify the trials according to pharmacological treatments, this has not been possible due to the notable differences between therapeutic approaches.

Authors should edit the text through the article, for example see line 183.

The document has been thoroughly reviewed for typos.

Reviewer 3 Report

This is a useful and informative review of the importance of the immune response in HER-2 breast cancer. The manuscript is well written, concise, and a valuable addition to the literature.

Minor suggestions:

1. The figures contained in the manuscript are helpful. Would consider an additional figure for Section 3.5 illustrating an M1 vs. M2 microenvironment.

2. In Section 3.5, would comment briefly - 1-2 sentences - on the impact of involution on promoting M2 polarization. 

Author Response

RESPONSE TO REVIEWER 3 COMMENTS

Comments and Suggestions for Authors

This is a useful and informative review of the importance of the immune response in HER-2 breast cancer. The manuscript is well written, concise, and a valuable addition to the literature.

Thank you for reviewing our work.

Minor suggestions:

  1. The figures contained in the manuscript are helpful. Would consider an additional figure for Section 3.5 illustrating an M1 vs. M2 microenvironment.

We very much appreciate your suggestion to improve our work. However, the initial aim of this manuscript is to review the role of TILs in the resistance of HER2-positive breast cancer patients. Thus, from our point of view, the relevance of the role of macrophages in these processes should be placed in a context under another perspective. In the same sense, we have chosen to summarize all the available information on each population of the immune system, both in the text and in Figure 1. For this reason and due to the scarce evidence, we believe that we cannot extract enough information to provide a figure on the specific role of M1 and M2 macrophages in this group of patients with HER2-positive breast cancer.

  1. In Section 3.5, would comment briefly - 1-2 sentences - on the impact of involution on promoting M2 polarization. 

In accordance with the reviewer's comments, we have included more extensive information on TAM repolarization. Therefore, we have tried to provide a more detailed explanation on this point, highlighting its promising impact on clinical practice. Unfortunately, there are few references available on this topic in HER2-positive patients and therefore we have not been able to expand this discussion as much as we would have wished.